# Phylogenetic conservation of bacterial responses to soil nitrogen addition across continents

Kazuo Isobe[1,2], Steven D. Allison [1,3], Banafshe Khalili[1], Adam C. Martiny [1,3] & Jennifer B.H. Martiny[1]

Soil microbial communities are intricately linked to ecosystem functioning such as nutrient cycling; therefore, a predictive understanding of how these communities respond to environmental changes is of great interest. Here, we test whether phylogenetic information can predict the response of bacterial taxa to nitrogen (N) addition. We analyze the composition of soil bacterial communities in 13 field experiments across 5 continents and find that the N response of bacteria is phylogenetically conserved at each location. Remarkably, the phylogenetic pattern of N responses is similar when merging data across locations. Thus, we can identify bacterial clades – the size of which are highly variable across the bacterial tree – that respond consistently to N addition across locations. Our findings suggest that a phylogenetic approach may be useful in predicting shifts in microbial community composition in the face of other environmental changes.

[1] Department of Ecology and Evolutionary Biology, University of California, Irvine, CA 92697, USA. [2] Graduate School of Agricultural and Life Sciences, The University of Tokyo, Tokyo 113-8657, Japan. [3] Department of Earth System Science, University of California, Irvine, CA 92697, USA. Correspondence and requests for materials should be addressed to K.I. (email: akisobe@mail.ecc.u-tokyo.ac.jp) or to J.B.H.M. (email: jmartiny@uci.edu)

I t is well established that environmental changes such as soil warming and nutrient addition alter the composition of bacterial communities[1]. However, the enormous diversity of soil bacteria precludes characterizing each individual bacterial taxon to predict how a community will respond to environmental perturbations. Recent studies suggest that microbial responses to environmental changes may be phylogenetically conserved across the tree of life[2–5] such that closely related bacterial taxa respond more similarly to a perturbation than those that are distantly related (Fig. 1a). This pattern exists despite horizontal gene transfer and rapid evolution among bacteria, which might break up the signal of vertical inherence and result in a random distribution of response across the phylogeny[6]. One can further consider the taxonomic or genetic resolution at which bacterial responses are phylogenetically conserved. For instance, soil bacteria within a phylum appeared to respond similarly to added water[2], whereas soil bacteria within a family tended to respond similarly to carbon addition[4], indicating that responses to water availability may be more deeply conserved than those to carbon availability. Together, these studies indicate that phylogenetic information about bacterial responses could be used to predict the response of uncharacterized but phylogenetically related taxa[1,7].

**Fig. 1** The conceptual framework for the study. Bacterial taxa respond either positively (blue) or negatively (red) to an environmental perturbation. **a** The responses might be phylogenetically conserved (left) or random (right). **b** The responses might be phylogenetically conserved at each location, but context dependent. In this illustration, the response of nine taxa in location A differ in location B. **c** The response of individual taxa might be consistent across locations (not be context dependent), but estimates of the depth of conservation might vary because of the other taxa present. Here, a positively responding (blue) clade appears very deeply conserved in location A, but is broken up into shallower clades in location B. **d** To test the context dependency of responses to a perturbation and to identify clades that respond consistently across locations, one can compare the responses at individual locations (left) to those observed when datasets are merged across locations (right)

Still, it remains unclear how well data on bacterial responses at one location predict bacterial responses at a different location – that is, whether the responses are context dependent. In particular, the response of individual taxa (or clades of taxa) might be contingent on the magnitude of perturbation applied as well as the history and baseline of the environment at each location (Fig. 1b). A taxon's response might further depend on biological interactions with other taxa present in the community[8], the composition of which varies from location to location. In addition, the nature of microbial sequence data means that a taxon's response to a perturbation is usually quantified relative to other taxa in the dataset[9]. Thus, the same response (e.g., a 10% reduction in absolute abundance) of a taxon in two different communities might appear to be either an increase or decrease in relative abundance.

Even if taxa are consistent in their responses across locations, differences in community composition might alter the genetic depth at which bacterial responses appear to be phylogenetically conserved. Specifically, the taxa present in a community (or abundant enough to be detected) might influence the extent to which a particular phylogenetic clade shows a consistent response (Fig. 1c). For instance, if a phylum is only represented by a handful of taxa that all respond similarly to the perturbation in one location, then one might extrapolate that other taxa within the same phylum will respond uniformly. However, sampling of additional taxa within that phylum from other locations might reveal finer scale variation. Hence, even if bacterial responses are conserved and not context dependent, community responses might still be unpredictable across locations.

Fortunately, the contribution of context dependence of bacterial responses can be tested by comparing the phylogenetic patterns of bacterial responses within individual locations to those assessed across multiple locations (Fig. 1d). If bacterial responses are context dependent, then the genetic depth at which the responses are conserved in the merged dataset would be shallower than at the individual locations. Alternatively, if the degree of conservation is similar in the merged and individual datasets, then we can identify specific clades that respond consistently to N addition.

Here, we test these ideas by investigating the response of soil bacteria to nitrogen (N) addition. Anthropogenic activity has dramatically modified N cycles of terrestrial ecosystems[10], and global N deposition is predicted to increase worldwide 50–100% from 2000 to 2030[11], further impacting global plant biodiversity and microbial functioning[12]. Numerous studies report that experimental N addition alters soil microbial community composition[1,13], and some have identified consistent changes at the community level as suggested by ordination analysis[14–16]. At the same time, specific comparisons of broad taxonomic levels (e.g., phylum/class) show inconsistent responses across studies.

For instance, the phylum Verrucomicrobia responded positively to N addition in cropland soils in China[17] and grassland soils in the United States[18], but responded negatively in alpine tundra soils in the United States[19]. Such conflicting results would seem to indicate that the bacterial N response is highly context dependent or alternatively, phylogenetically conserved at a shallower depth.

We re-analyzed publicly available data from soil N addition experiments to ask: (1) Is the response of soil bacteria to added N phylogenetically conserved at each individual location? (2) Are the N responses of particular taxa (or clades of taxa) context dependent or consistent across locations? Finally, (3) can we identify specific taxonomic groups that respond to N addition consistently across locations? We collated 16S rRNA gene sequences from soil N addition experiments from various biomes with different levels of N addition (Table 1, Supplementary Fig. 1, and Supplementary Table 1), comparing bacterial community composition of treatment versus control plots at 13 locations across 5 continents. To address the first question, we considered not only if the bacterial response is non-randomly distributed across the phylogenetic tree, but also the degree to which the response is conserved (i.e., the genetic depth at which descendent bacterial taxa show a similar response to N addition). To address the second and third questions, we merged datasets of widespread (present at 5 locations or more) OTUs (operational taxonomic units defined at >97% sequence similarity) and tested the robustness of phylogenetic conservation of bacterial N responses. This broader definition of taxa (rather than exact sequence variants as in ref. [20]) allowed us to compare the same taxa across many locations, which was key to our analysis. We find that bacterial N responses are phylogenetically conserved and that the context of the particular location and experiment does not overshadow the phylogenetic signal. This allows us to identify phylogenetic clades that respond consistently, positively or negatively, across all locations. Such robust phylogenetic signals offer specific predictions about how the composition of a soil bacterial community at other locations will respond to N addition.

## Results

**Phylogenetic conservation of bacterial N responses.** We first quantified the response of bacterial OTUs to N addition at each individual location (Supplementary Fig. 2). The responses were phylogenetically conserved at all locations, as supported by several analytical approaches. Consensus clades (>90% of OTUs responding in the same direction) identified by the consenTRAIT algorithm[21] were dispersed throughout the phylogenetic tree at each location (Supplementary Fig. 3). The mean genetic depth ($\tau_D$) for both positive and negative responses ranged from 0.016 to 0.020 (average $\tau_D = 0.018$ or an average sequence dissimilarity in the 16S rRNA gene amplicon of ~3.6% among OTUs, Table 2).

**Table 1 Characteristics of study locations**

| Locations | Habitat (country) | No. replicate plots | Level/type of N addition | Number of OTUs |
|---|---|---|---|---|
| Carey_1[31] | Clipped grassland (USA) | 8 | 45 kg-N ha$^{-1}$ yr$^{-1}$ of $NH_4NO_3$ | 1851 |
| Carey_2[31] | Non-clipped grassland (USA) | 8 | 45 kg-N ha$^{-1}$ yr$^{-1}$ of $NH_4NO_3$ | 2110 |
| Leff_1[16] | Grasslands (Switzerland) | 3, 6[a] | 100 kg-N ha$^{-1}$ yr$^{-1}$ of urea | 2089 |
| Leff_2[16] | Grassland (Australia) | 3, 6[a] | 100 kg-N ha$^{-1}$ yr$^{-1}$ of urea | 1263 |
| Leff_3[16] | Grassland (South Africa) | 3, 6[a] | 100 kg-N ha$^{-1}$ yr$^{-1}$ of urea | 1094 |
| Leff_4[16] | Grassland (South Africa) | 3, 6[a] | 100 kg-N ha$^{-1}$ yr$^{-1}$ of urea | 1624 |
| Leff_5[16] | Grassland (South Africa) | 3, 6[a] | 100 kg-N ha$^{-1}$ yr$^{-1}$ of urea | 1444 |
| Leff_6[16] | Grassland (Australia) | 3, 6[a] | 100 kg-N ha$^{-1}$ yr$^{-1}$ of urea | 1177 |
| Li_1[32] | Bamboo forest (China) | 3 | 30, 60, 90 kg-N ha$^{-1}$ yr$^{-1}$ of $NH_4NO_3$ | 3303 |
| Li_2[32] | Bamboo forest (China) | 3 | 30, 60, 90 kg-N ha$^{-1}$ yr$^{-1}$ of $NH_4NO_3$ | 2287 |
| McHugh[30] | Grassland (USA) | 5 | 8 kg-N ha$^{-1}$ yr$^{-1}$ of $NH_4NO_3$ | 1198 |
| OBrien[18] | Cropland (USA) | 3 | 67 kgN ha$^{-1}$ yr$^{-1}$ of urea | 8612 |
| Wang[33] | Cropland (China) | 3 | 160 kg-N ha$^{-1}$ yr$^{-1}$ of urea | 2953 |

[a]3 N addition plots and 6 control plots

**Table 2 Mean genetic depth ($\tau_D$) of consensus clades as calculated with the consenTRAIT algorithm**

| Locations | Positive response | Negative response |
|---|---|---|
| Carey_1[31] | **0.017** | 0.018 |
| Carey_2[31] | **0.018** | 0.018 |
| Leff_1[16] | **0.020** | 0.018 |
| Leff_2[16] | **0.020** | 0.019 |
| Leff_3[16] | **0.018** | 0.019 |
| Leff_4[16] | 0.018 | 0.017 |
| Leff_5[16] | **0.017** | 0.018 |
| Leff_6[16] | 0.016 | **0.020** |
| Li_1[32] | **0.019** | 0.018 |
| Li_2[32] | **0.017** | 0.018 |
| McHugh[30] | **0.020** | 0.020 |
| OBrien[18] | **0.018** | 0.017 |
| Wang[33] | **0.019** | 0.018 |
| Mean of each location | 0.018 | 0.018 |
| Merging locations | **0.017** | **0.017** |

Bold indicates that the response is significantly associated with phylogeny (permutation test; $P$ <0.05). Consensus clades are the phylogenetic clades in which >90% of the descendant OTUs show the same direction of response.

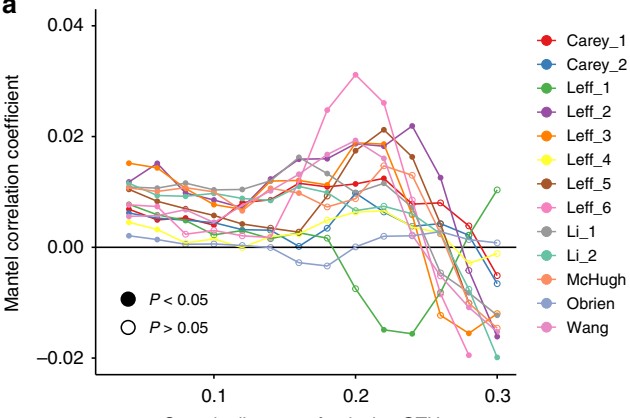

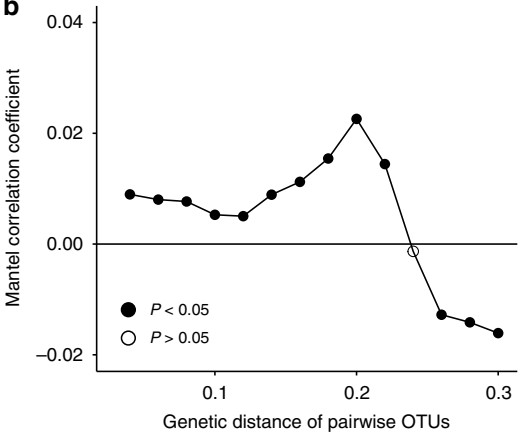

**Fig. 2** Mantel correlograms showing the difference in response versus genetic distance of pairwise OTUs. **a** Response of bacterial OTUs to N addition at each location and **b** the averaged response across locations of widespread OTUs (i.e., present at five locations or more) ware analyzed. Filled symbols represent genetic distance classes in which the autocorrelation coefficient significantly differs from zero

For all locations except one, this depth was greater than expected given a randomized distribution of responses (permutation test; all $P < 0.05$ except Leff_4). The $D$-test of Fritz and Purvis[22] also confirmed that the N responses were phylogenetically non-randomly distributed at every location (permutation test; $P < 0.05$, Supplementary Table 2). These results were largely robust to the phylogenetic reconstruction method (neighbor-joining or maximum likelihood), as expected from the high correlation between the trees for all datasets (Supplementary Table 3).

Beyond the direction of the responses, the magnitude of N responses was also phylogenetically structured. A phylogenetic correlogram analysis[23] showed that the similarity in response (considering the response as continuous data) of any pair of OTUs was significantly positively correlated with the pair's genetic relatedness at all locations; this correlation was consistently positive when pairs of OTUs were at least 6% genetically dissimilar (for Leff_4, Leff_6, and O'Brien) and up to 25% dissimilar (for Leff_2 and Li_2) (Fig. 2a). At very high genetic distances the correlations become negative, capturing opposing responses across deeper clades.

Merging datasets across locations, we observed similar phylogenetic patterns of the N responses of the widespread OTUs (Fig. 3). These OTUs (present in at least 5 locations) accounted for 68–93% of the sequences at any one location (Supplementary Fig. 4). The genetic depth of consensus clades was similar to that observed at individual locations (mean $\tau_D = 0.018$) with a value of $\tau_D = 0.017$ (permutation test; $P < 0.05$) for both positive and negative responses (Table 2). The $D$-test also confirmed that responses were phylogenetically non-randomly distributed (permutation test; $P < 0.05$; Table S2). Finally, the phylogenetic correlogram analysis revealed that the similarity in the magnitude of the N response for any pair of widespread OTUs was significant for OTU pairs up to 22% dissimilar (Fig. 2b).

We next identified taxonomic groups that showed a consistent response to N addition across locations (Fig. 4). At the broadest level, OTUs within the phylum Actinobacteria tended to respond positively, whereas OTUs within the phyla Acidobacteria, Planctomycetes, Bacteroidetes, and Gemmatimonadetes, and the classes Betaproteobacteria and Deltaproteobacteria, tended to respond negatively (two-tailed exact test; all $P < 0.05$). However, broader taxonomic patterns such as these often obscured patterns within

taxonomic groups. For instance, the class Gammaproteobacteria did not respond consistently in one direction, but the genus *Aquicela* within Gammaproteobacteria responded positively (two-tailed exact test; $P = 0.05$) (Fig. 4). Similarly, the class Acidobacteria was not consistent in its overall response, but an undefined genus within the family *Acidobacteriaceae* (subgroup 1) responded negatively. We also detected one case in which a broad taxonomic clade responded significantly one way, but a finer group within that clade responded in the opposite direction; the order Sphingobacteriales (phylum Bacteriodetes) responded significantly negatively to N addition, but the relatively small family *Sphingobacteriaceae* responded strongly positively (10 out of 11 OTUs).

## Discussion

While a response to N addition is not a trait itself, the degree to which a response is conserved can give clues to the underlying traits – those that determine a taxon's change in abundance in response to environmental change (so-called response traits[1,24,25]). Bacterial responses to N addition were phylogenetically conserved at a genetic depth ($\tau_D$) of 0.018 within locations and 0.017 across locations. This depth corresponds to a 3.4–3.6% divergence in the 16S rRNA gene amplicon, or approximately the level of a bacterial genus[26]. A previous study of leaf litter bacteria found that the positive response to N addition was conserved at a similar depth ($\tau_D = 0.020$)[5]. This depth is

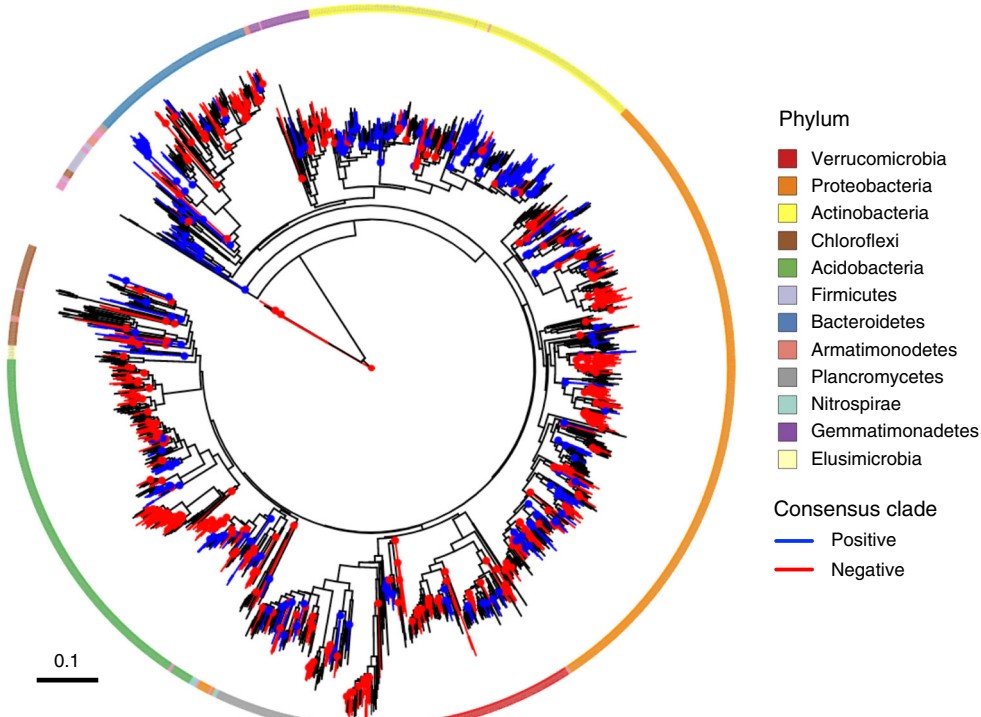

**Fig. 3** Phylogenetic distribution of the averaged response across locations of widespread OTUs (i.e., present at 5 locations or more). Colored nodes and lineages show the consensus clades in which >90% of the descendant OTUs show the same direction of response (blue: positive response, red: negative response). Note that all OTUs in the tree do respond to N, but only consensus clades are colored for clarity. The outer ring shows the phylum-level taxonomy of OTUs determined using the RDP classifier trained on the SILVA database

more conserved than bacterial traits such as the ability to produce particular extracellular enzymes ($\tau_D < 0.010$)[27] or the use of simple carbon compounds for growth ($\tau_D < 0.010$)[21] but less conserved than complex metabolic pathways such as oxygenic photosynthesis ($\tau_D = 0.101$) or aerobic methane oxidation ($\tau_D = 0.046$)[21].

Although many factors influence the phylogenetic distribution of traits, the genetic complexity of a trait (quantified as the number of genes in a pathway or the chemical complexity of carbon compounds targeted) seems to be positively correlated with its depth of conservation[21]. By this relationship, the direction of the N response would appear to be intermediate in its genetic complexity. We hypothesize that a bacterium's N response is not encoded by one gene or metabolic pathway. Indeed, a recent study of publicly available bacterial genomes noted that the pathway for ammonia assimilation was nearly ubiquitous (93% of genomes) in bacteria[28] such that the presence of this pathway cannot help distinguish differences in the responses observed here. Instead, the response likely depends on a mixture of nutrient transporters, the ability to grow quickly when N availability suddenly increases, as well as subtle differences in many pathways that alter the nutrient requirements of the cell. N addition might also have indirect effects on bacterial abundance through changes in plant communities and soil and litter chemistry[29]. Further, N addition can influence edaphic properties such as soil pH, although the studies included here noted only very small changes in soil pH with N addition[16,18,30–33]. Overall, this complexity makes it difficult to unravel the mechanisms, let alone identify the bacterial traits, underlying N responses.

The depth of consensus clades considers the direction of the N response, but we also found that the similarity in the magnitude of N responses between OTUs was positively correlated with their genetic similarity. The phylogenetic depth of this correlation varied by location, perhaps due to variation in the tree topology of the community present. In the merged dataset, however, OTUs up to 22% divergent in the 16S rRNA gene amplicon tended to respond with similar magnitude (Fig. 2). As expected, this metric of conservation depth is less conservative than consenTRAIT's metric, as the latter only considers consensus clades rather than an overall trend in direction.

Although the average depth of the N responses was highly consistent across locations, it is important to recognize that the genetic depth of individual clades across the bacterial phylogeny was highly variable (Supplementary Fig. 5). Such variation is expected, because clades evolve independently. The relatively shallow average depth of N responses indicates that the traits underlying N responses are generally evolutionarily labile, but – whether by variation in chance mutations, historical selection pressures, or interactions with the genomic background – there are also clades where N response is deeply conserved.

This variation in the depth at which the N response is conserved is key to using this framework for predictive purposes. Predictions about how a new, unobserved community will respond to N addition will be improved if specific clades that respond to N addition consistently are identified rather than applying an average prediction based on one taxonomic level or an average clade depth. We suspect that this variation in conservation underlies many of the conflicting results among previous studies of N addition experiments. Indeed, most of these studies conduct a taxonomy-based analysis and compare the magnitude of responses among a particular taxonomic level (e.g., phyla)[14–16]. For example, Leff et al.[16] combined data across 25 grassland locations (including datasets reanalyzed here) and found that, on average, some bacterial phyla responded consistently positively (Actinobacteria, class Alphaproteobacteria, and class Gammaproteobacteria) or negatively (Acidobacteria, Plancromycetes, and class Deltaproteobacteria) among the locations, whereas other phyla

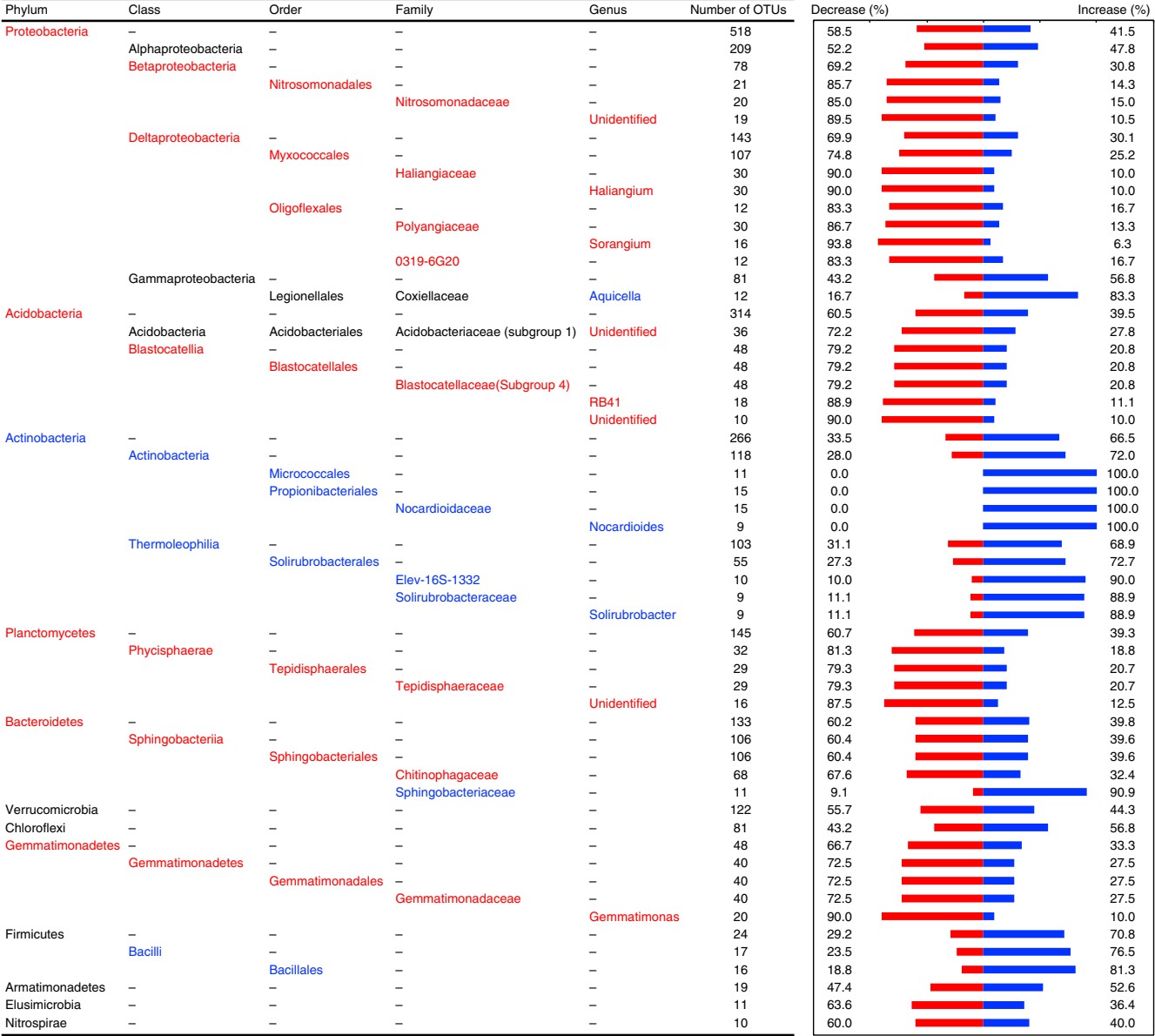

**Fig. 4** Taxonomic groupings whose response to N addition was positive or negative. The response significantly more positive (blue text) or negative (red text) than expected by chance (two-tailed exact test; $P < 0.05$), based on the response ratios of each OTU calculated in DESeq2, are shown. We also include all phyla (and classes for Proteobacteria) even if they were not significant (black text) for comparison. The percent of OTUs responding positively (blue) or negatively (red) are plotted in the bar graph to the right. Higher taxonomic levels are listed in black (e.g., the order *Legionellales*) when only the lower levels are significant

did not respond. In contrast, our study and others[15,17,18,31,34] found different responses of the same phyla.

Our results suggest two reasons for these discrepancies. First, N responses are conserved on average at a much shallower depth than phyla, and second, some phyla, such as Verrucomicrobia, Chloroflexi, and Armatimonadetes (Fig. 4), may be less consistent than others in their response to N addition. Thus, reporting results in a phylogenetic context and developing better analytical methodologies to identify clades driving community shifts[35] will aid in direct comparison of microbial community studies.

Finally, it is important to note that most of the locations studied (9 out of 13) were grasslands, which limits the variation in bacterial composition and environmental factors observed[36]. Thus, including experiments from a wider range of soil biomes would help to improve the extent to which these results can be applied to other systems. Additional sampling would also increase

the degree of phylogenetic representation across the bacterial tree of life and thereby, improve our ability to predict the N responses of particular clades.

Emerging evidence indicates the response of bacteria to other disturbances such as water addition[2,3], carbon addition[4], and drought[5] are also phylogenetically conserved. Further, some metabolic pathways and process rates (including N assimilation) seem to be phylogenetically patterned as well[21,37]. While the extent to which these patterns are context dependent remains to be tested, they suggest that phylogenetic information may be useful in predicting how future environmental changes will alter microbial communities and their functioning.

## Methods

**Study inclusion criteria**. We searched for published studies that assessed the compositional shift of soil bacterial/archaeal communities (based on 16S rRNA gene

sequences) by the experimental manipulation of N addition that met the following criteria: (1) published before 10 Sep 2017, (2) included at least three replicates for N addition and control (non-addition) plots, (3) used high-throughput amplicon sequencing covering the V4 region of 16S rRNA genes in bacteria/archaea that can be amplified with the primers 515F/806R, and (4) sampled from surface soil. We identified 15 published studies that fit our criteria but excluded 9 studies because the raw sequence datasets and/or accompanying metadata were not publicly deposited or otherwise obtainable. Table 1 describes the overview of 13 experimental locations from 6 published studies that were included in our analysis. The locations were spread over 5 continents (Supplementary Fig. 1). The overall procedures of this study are summarized in Supplementary Fig. 6.

**Sequence processing.** We compiled the 16S rRNA sequences from all experimental locations. To ensure consistent comparisons across studies, we re-assigned sequences to OTUs. Specifically, fastq-formatted raw sequence files and the associated metadata were either shared by the original authors or downloaded from the NCBI Sequence Read Archive (SRA), the EMBL European Nucleotide Archive (ENA), or the Metagenomics RAST Server (MG-RAST). The sequence datasets, all of which were sequenced on the Illumina Miseq platform, were demultiplexed within the QIIME pipeline (ver. 1.9.1[38]) with the split_libraries_fastq.py script. Primer sequences, when included, were removed with the fastq (https://github.com/dcjones/fastq-tools) and FASTX (http://hannonlab.cshl.edu/fastx_toolkit/) toolkits. In the case of one dataset[18], we combined the sequences from multiple subsamples by plot. In the case of another dataset[32], we combined the sequences from the plots receiving different N amounts.

The UPARSE pipeline[39] was used to merge the paired-end sequences, conduct quality filtering, and cluster the sequences into OTUs. A minimum overlap of 20 bp was set for merging the sequences. A maximum per sequence expected error frequency value of 0.5 was set for quality-filtering the sequences, and singleton sequences were removed. Paired-end sequences from all experimental locations were merged and clustered into OTUs at >97% sequence similarity, and chimeric sequences were further removed using UCHIME algorithm with gold database (https://www.drive5.com/uchime/gold.fa). Taxonomy of the representative sequence of each OTU was assigned within QIIME using the RDP classifier[40] at 80% confidence threshold trained on the SILVA database (version 128, https://www.arb-silva.de/download/archive/qiime/). OTUs assigned as chloroplasts or mitochondria, or unassigned at Kingdom level were removed, yielding 4210–12,387 OTUs per location. Only 79 OTUs were assigned to Archaea; although we kept them in the analysis, we discuss the results in terms of bacterial responses because Archaea made up only 1.6% of all sequences.

**An OTU's N response at each location.** We first quantified the response of the bacterial OTUs (including several archaeal OTUs) to N addition at each location. To do this, we used the DESeq2 package[41] in the platform of phyloseq package[42] in the R environment (http://www.R-project.org). We used DESeq2 in a limited way that differs from its typical use for RNAseq data. Specifically, we used it to (1) normalize the sequence counts by sample within a location by replacing the original counts with variance stabilized counts and (2) calculate the $\log_2$-fold ratio of averaged relative abundance in N addition plots relative to control plots for each taxon. Before calculating the response ratios, we first removed rare OTUs present in less than half of all the plots within a location using the metagMisc package (https://github.com/vmikk/metagMisc)(Supplementary Table 1), as the response ratios of low occupancy, low abundance taxa would be subject to a high degree of noise. Note that we did not use DESeq2 to test for statistical significance (as the program is often used), but exported the normalized $\log_2$-fold ratios for further analyses below.

**Phylogenetic conservation of N responses at each location.** To assess whether the N response was phylogenetically conserved at an individual location, representative sequences of each OTU (the most abundant sequence within each OTU) were aligned using the DECIPHER package[43]. A neighbor-joining (NJ) phylogenetic tree was inferred with bootstrap analysis (100 replicates) using the phangorn package[44]. We then applied three approaches (consenTRAIT analysis, D-test of Fritz and Purvis, and phylogenetic correlogram) to test whether an OTU's response to N addition was related to the bacterial phylogeny. Each approach has different strengths and limitations.

For the consenTRAIT analysis, we followed the methods of Martiny et al.[21]. The consenTRAIT algorithm[21] identifies phylogenetic clades in which a trait is conserved (consensus clades) and calculates the average genetic depth of those clades. This approach only considers binary traits (here, whether the N response is positive or negative). A positive ($\log_2$-fold ratio >0) or negative ($\log_2$-fold ratio <0) response was assigned for each OTU on the NJ phylogenetic tree based on the $\log_2$-fold ratio exported from DESeq2. The tree was traversed from the root to the tips, recording the deepest nodes where >90% of the descending OTUs (tips) shared the same directional response (a 'consensus' clade). The genetic depth (average distance of the node to its descending tips) and size (total number of the descending tips) of each consensus clade was calculated. The genetic depth of clades with a single descending tip (OTU) was calculated as half the branch length to the nearest neighbor as previously recommended[21]. Finally, the mean genetic depth, $\tau_D$, of the consensus clades sharing either positive or negative responses was calculated. To assess the statistical significance of phylogenetic conservation of N

responses, simulated $\tau_D$ values were calculated by randomizing the responses among the tips 1000 times. The probability of phylogenetic conservation (non-randomness) of the traits was calculated as the fraction of simulated $\tau_D$ values that were greater than or equal to the observed $\tau_D$.

We used NJ trees for the consenTRAIT analysis, because the genetic scale of these trees roughly represents sequence dissimilarity. However, to consider whether our results were robust to the phylogenetic reconstruction method, we tested for a correlation between NJ and maximum likelihood (ML) trees for each dataset. We also tested whether the N responses were significantly associated with the ML phylogeny using the consenTRAIT metric. A ML tree with 100 bootstrap replications was constructed with RAxML v8.0, using the GTR + Gamma distribution model[45].

The D-test of Fritz and Purvis[22] has been widely applied to test for a phylogenetic signal of macroorganism traits. Like consenTRAIT, it only considers binary traits. However, the D-test is less strict than consenTRAIT in testing for a phylogenetic signal, because it considers clades with mixed responses. We estimated the phylogenetic dispersion (D) of the N response (positive or negative) using the caper package[46]. We permuted (1000 times) the tips on the phylogenetic trees based on a random pattern of evolution. The statistical significance of the observed pattern was assessed by calculating the fraction of simulated D values that were smaller than or equal to the observed D value.

The third approach, phylogenetic correlogram analysis[23], considers continuous traits (here, the magnitude and direction of an OTU's response). However, because the correlations are computed within discretized distance classes, the more unequally OTUs are distributed on a phylogenetic tree, the weaker the test's statistical power. We used the approach to test the phylogenetic scale at which the difference in N response values ($\log_2$-fold ratios) between any pair of OTUs is correlated with the phylogenetic distance between them, following the method of Amend et al.[5]. Phylogenetic distance between OTUs was calculated as the proportion of nucleotide sites that differ between their aligned 16S rRNA sequences using the DECIPHER package[43]. An autocorrelogram was plotted along a continuous scale of phylogenetic distance. The statistical significance of the correlation at each distance interval was calculated based on 999 permutations of the responses using the vegan package[47].

**Phylogenetic conservation of N responses across locations.** To assess whether the N responses were context dependent, we identified 1726 OTUs that were present in five or more experimental locations. (Note these OTUs were from the pool of non-rare OTUs in each location; Supplementary Table 1). For each of these widespread OTUs, we averaged the response values across locations. This procedure treats the results of each location equally, regardless of differences in methods and sequencing effort among location. Unlike for most diversity metrics, the response ratio parameter that we estimate for each taxon in a location should not be biased by sequencing effort, although it will presumably get more accurate with more sequencing. We then created a NJ tree of these widespread OTUs and performed the consenTRAIT, D-test, and correlogram analyses as above with this merged dataset.

Because the N responses of widespread OTUs were significantly phylogenetically conserved, we identified the taxonomy of clades whose response to N addition was significantly more positive or negative than expected by chance. Based on the RDP classifications, we calculated the number of OTUs that had a positive or negative response at each phylum, class, order, family or genus level. We performed a two-tailed exact test[48] against the equal distribution of positive and negative responses within each taxonomic group.

**Reporting summary.** Further information on research design is available in the Nature Research Reporting Summary linked to this article.

## Data availability

Raw sequence files and the associated metadata are available from the original studies, of which sources were summarized in Supplementary Table 1. Our reanalysis of the datasets is available at https://github.com/kazuo-isobe/phylogenetic-conservation-N-addition and from the authors upon request.

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

## Acknowledgements

We thank the authors whose data contributed to this analysis. We also thank C. Weihe, S. Glassman, A. Chase, K. Walters, L. Vivanco, S. Finks, C. Rodriguez for helpful comments on the manuscript. This work was supported by the National Science Foundation (DEB-1457160) and the U.S. Department of Energy, Office of Science, Office of Biological and Environmental Research (DE-SC0016410). KI was supported by JSPS Overseas Research Fellowships (995308) and KAKENHI (18H02233).

## Author contributions

K.I., B.K., and J.B.H.M. analyzed the data. K.I., S.D.A., A.C.M., and J.B.H.M. conceived and designed the analyses. K.I. and J.B.H.M. wrote the paper with editorial contributions from S.D.A. and A.C.M.

## Additional information

**Competing interests:** The authors declare no competing interests.

