## [Peer Review File · Nature Communications]

Reviewers' Comments:

Reviewer #1:

Remarks to the Author:

This work examines the concerted responses of soil bacterial taxa to nitrogen addition across six published studies that each compared a nitrogen amendment to an untreated control. The major finding is that, for many taxa, a taxon's positive or negative response to nitrogen addition is largely consistent across studies and within its clade. This study asserts that these conserved responses will be useful to predict the most likely responders to nitrogen additions by considering the composition of the pre-N addition community. Overall, the study is interesting, and stops just short of moving forward from the observation.

I make two major comments and a few minor ones. Thank you for the opportunity to read this piece. I hope that my comments are helpful to the authors.

Major comments

1. Bench marking sampling effort. The introduction discusses differences in sampling effort across studies (paragraph starting Line 52) and how this may bias the ability to extrapolate and interpret predictions. It is suggested that understanding the phylogenetic conservation of responses will help to overcome limitations of under-sampling. Despite this emphasis on issues with sampling in the introduction, there are two steps in the methods (pertaining to sampling) that could be benchmarked to assess the influences of those steps on results. These steps should be assessed to determine whether they alter results or interpretation.

a. First, study does not standardize for observational effort (here, sequencing depth) across the different studies included. The study uses DESeq2 for assessing log₂ changes without prior standardization to an equal sequencing depth across samples. This will allow for some rare members to be observed in samples that have better observational effort (deeper sequencing), and may influence results. I have not noticed any meta-analyses that have used DESeq2 to standardize across studies (that generated data from different sequencers, labs, DNA extraction protocols, etc) but some benchmarking will help the field to understand DESeq2 utility for meta-analyses in general and for consistency in N responses in particular. I suggest performing a direct comparison of resulting log₂-changes across data subsampled to an equal observational effort and data that has not been.

b. Second, the study omits "noise" by removing taxa that were detected in less than half of the plots per study (L241). This step may allow for some responsive taxa with lower occupancy to be not considered across studies, which may also influence results. It will also be more likely to filter rare taxa, which are statistically expected to have lower occupancy. However, including all observed taxa is one of the advantages of DESeq2 – that one doesn't have to "throw out" high-quality data despite differences in sequencing effort across samples. What is the benefit of this step for the log₂-change results and how does omitting it change outcomes?

c. I also wonder about the consequences of removing taxa with less than 50% occupancy prior to the DESeq2 log₂ change analysis – my understanding was that the shrinkage approach of DESeq2 requires all data, including the noisy points, to best estimate the variance structure before calculating log₂ fold changes?

d. Information can be added to the main text, like QC sequencing depth across studies (in addition to the total number of OTUs observed, number and taxonomy of OTUs excluded at the occupancy filter step, to help readers interpret the range and variation in the studies included and its consequences for cross-study comparisons.

2. Moving towards mechanisms. The study ends with a conclusion of consistent N responses across studies and conserved within some major lineages. It is a brief piece that stops short of investigating the "why". The next step is to ask what functions these lineages have that support their consistent

responses. The article would be enhanced if there were some discussion, or, better, some investigation, of the functional genes related to nitrogen responses/cycling that are conserved within the responsive lineages (L117). There are genomic resources available for many of the responsive clades – the authors could consider some representative comparative genomic cases, perhaps of the most phylogenetically conserved and maximally responsive lineages, to explore their genomes for conserved functional genes that would explain their responses. They could also test their complexity hypothesis of related genes (discussion starting L150).

Minor comments

L86 – a bit unclear as to whether the entire set of analyses were performed at the 97% OTU level or just the cross-location analyses? I'm not sure if the exact sequence variants comment is needed if these are not ever used.

Were there any taxon/lineage responses conserved within nitrogen level or type (as reported in Table 1)? One hypothesis is that that type of nitrogen would influence taxa that directly respond to particular substrate but potentially not indirectly responsive taxa.

Though Figure 1 provides a nice visualization for the types of patterns that could be observed, some aspects of it are unclear. Panel A dendrograms demonstrate potential different patterns (under and over dispersion) within a particular location, which is not directly addressed in the main text. Panel B is comparing phylogenies across two locations, and also builds into the protocol in Panel D and so may not be needed as a separate panel. Panel C is representing the sampling issue, but is not directly addressed in the paper with analysis (though bench marking as described above would help this). Panel D summarizes the study at hand. Overall, I think this figure could be removed. If the authors want to keep it, please clarify these parts and consider moving it to supporting materials.

Figure 2A – the Green trace (Leff_1?) is an obvious outlier. Why?

Figure 3 – the authors describe lineages that have conserved responses at different resolutions, but this tree is too crowded to discern some of those specific discussion points. I'm not sure that the tree is helpful.

Tables 1 and 2 – I wonder if these could instead be represented as figures? For example, as bars centered around zero, with increases presented as positive bars, and decreases as negative? As it stands, it is hard to digest the range and variability in the responses by reading such a large table, and it is also hard to "ignore" the non-significant changes that are presented in regular (not bold) font. The Table legends should clearly state how the results were derived. I think that Table 1 is related to the output from DESeq2, but then I am confused as to how the percentages in change are calculated, because I thought that the test considered log₂-fold change, so shouldn't the change be represented as log₂-fold rather than %? Similar comment for Table 2. If results are not significant, I suggest removing values and just stating "n.s." Has some kind of multiple comparison adjustment been performed?

For Table 2 - it strikes me how there is not much of a range in genetic depth (TD), even among statistically insignificant values, though the original 2013 paper suggests that ~0.17 is a relatively high degree of phylogenetic conservation – it is so high that it is not within the range of the data shown therein (Fig 2). It would be interesting to see the distribution of these values – is there some expected range or variability of TD given the number of branches on the phylogeny?

Fig S4 – it looks like the Wang sample is missing?

Throughout, I was wondering if the percent change/divergence was referring to the full 16S rRNA gene or the amplicon (e.g., L 99, L 140, L162 and elsewhere).

L203 Inclusion criteria. This study has focused on native ecosystems like grasslands and bamboo forest, but is the N response consistent and conserved also in agricultural systems? Inclusion of agricultural manipulations would increase the study size and, potentially, the generality of the conclusions.

L 228 singleton sequences or OTUs? I interpret this as sequences.

A computing workflow (including sequence processing through to statistics in R) should accompany the work so that others can use and replicate the analysis. The metadata curated for this particular meta-analysis should be made available, as should the merged OTU table and the representative OTU sequences used to make the phylogenies because these are needed to reproduce the analyses (L315).

An interesting aspect of this work is that it is considering and analyzing N responses as if an increase or decrease after N amendment were in itself a particular microbial trait, but is it? As a contrast, nitrogen fixation is a bacterial trait encoded by a particular set of functional genes. It would enhance the work to include a brief discussion of responses v. traits, their overlap, and their distinctions, and how to use both to move forward toward predictions. If the authors include an analysis of functional genes conserved for N utilization within responsive lineages, this would allow them to potentially directly link responses to genetically-encoded traits.

Reviewer #2:

Remarks to the Author:

The main idea of this manuscript, i.e. to analyse whether soil bacterial responses to nitrogen additions are phylogenetically conserved, is relevant and timely. Results are based on the re-analysis of 13 experimental locations (from 6 published studies) that are distributed in five continents, and are thus potentially robust. The manuscript is well presented and easy to follow, but lacks several formal aspects that are required for a proper evaluation.

The authors present a conceptual figure to help follow the line of argument developed in the introduction. I generally understand (and like) the idea of having this figure, but it has several unclear elements and, so at the end it generates confusion. I particularly don't understand how local and sampling effects are represented in scenarios B to D. The authors should also justify why they exclude from their framework and several subsequent analyses those taxa that are not responsive to N additions. The lack of response in this case can be indeed very informative, and all locations and phyla contain many non-responsive OTUs (according to Fig. S2). Linked to this comment, I suggest that the magnitude of phylogenetic conservatism is tested after coding bacterial responses to N addition as a continuous trait (direction and magnitude of change, including 'no change' values) instead of coding the response as binary trait (positive vs negative) (and using e.g. Blomberg's K instead of D statistics). More information is needed in the materials and methods section to allow an adequate evaluation and future reproducibility of the study. This point is particularly relevant as regards two main questions. First, sequencing depth and OTU table construction: it is unclear whether the authors subsampled sequences at the same level in all studies and whether OTU tables represent number of counts or relative abundance. It is essential to justify this issue as most phylogenetic analyses are based on the positive (or negative) OTU response to N addition. In case OTU tables reflect relative abundances, the increase in some OTUs is necessarily accompanied by the decrease in other OTUs. Second, more details and analyses are needed as regards the phylogenetic reconstruction. Using short reads to

reconstruct phylogenies has several drawbacks that remarkably reduce tree reproducibility (often leading e.g. to many misplaced sequences even at the phylum level). I'd suggest that the authors recalculate all phylogenetic signals in several independently built trees to show that results are reproducible. When checking the consistency of their results, I'd suggest using the more robust maximum likelihood (instead of neighbour joining) trees. Please, find more details below on information that should be added to materials and methods.

Finally, the argumental flow in the introduction is clear and reasonable. However, many statements are not properly referenced based on published literature. Please, find some suggestions below.

Other comments

L20. five continents according to the main text and fig. S1

L26. Typo (add a full stop after 'location')

L39-41. This would be a good place to explicitly introduce the differential depth of conservation of bacterial traits, that you later analyse and discuss

L43-59. Please, support your arguments about local effects on published literature

L66-76. I'd move the paragraph on N addition to the first place in the introduction (this is a matter of taste, but I find it out of place here)

L75-76. Alternatively, could it indicate that the response is phylogenetically conserved at shallower clade depths?

L204-211. It would be a good idea represent the selection in a Prisma Flow Diagram in Supplementary Information (not necessary, but informative)

L215-234. The authors should make an effort to provide more detailed information. I'd suggest:

For published studies: sequence availability, sequencing technology (from L222 I understand that not all studies used Illumina, but I'm not sure), sequencing depth, treatment of different levels of N addition (did you somehow differentiate between N dose?), how did you treat replicated plots within locations?

For sequence processing: tool used for filtering chimera, reference SILVA database, sequencing depth used for OTU construction (did you standardize across studies?), content of OTU tables (do they reflect number of counts or relative abundance?)

For phylogeny reconstruction: please, give the phylogenetic trees and further details (e.g. did you align or remove hypervariable regions?)

L256. Could you add a third level (i.e. no response)? Otherwise, I'd change 'discrete' to 'binary'.

Table 1. What do asterisks mean in the column 'Replicate of plots'?

Fig1. Please, indicate in the caption what do left and right trees represent in scenarios B and C. How are locations represented in B. Also, I don't find scenario D very informative.

Fig2A. Maybe if dots were smaller, the figure would be easier to read

Table S1. What does the 'Mean' reflect in this case?

Figure S3. Could you make coloured bars showing the phylum thicker?

Kind regards,
Marta Goberna

Reviewer #3:

Remarks to the Author:

The authors investigated the response of bacterial taxa to nitrogen addition based on meta-analysis with phylogenetic information, which could provide a new insight to understand how soil microbes respond to environmental changes. The authors also identified specific taxa, which respond to N addition consistently across locations. The topic is interesting, and the manuscript is generally well written. However, there are some confusing points to make the results spurious.

(1) As we know, the duration and intensity of N addition will affect the extent of microbial response. In

Table 1, the authors listed the level/type of N addition in all the 13 sampling sites and the amount of N addition ranges from 8 to 160 kg N/ha/yr. This is a wide range of N addition, which can cause different levels of response. For the taxa with relatively wide ecological niche, they may not respond to 8 kg N/ha/yr N addition but respond to 160 kg N/ha/yr N addition. Is it acceptable to explore the microbial response to N addition without considering ~20-fold difference in N addition amount?

Besides, the duration and the timing of N addition is also important to understand microbial response to environment perturbations, but I can't find the related information.

(2) Another important point needed to be considered is the N nutrient status in the original sampling sites. If the sampling site is N-limited, we may detect more response, while if the sampling site is N-saturated, the response might be different. The lack of the ambient N condition in soils and ambient N deposition in each sampling site makes it hard to understand the findings in this study.

(3) The "global scale" is a little exaggerated as there are only 13 sampling sites included in the study. I feel 13 sites are not enough to produce robust results using meta-analysis.

(4) line 26, should be "across location. Our findings".

(5) line 78-81, question 2 and question 3 seem a little repeated.

(6) line 94-95, Figure S2 showed the changes of bacterial composition to N addition in phylum level rather than in OTU level.

(7) line 269-272. D-test of Fritz and Purvis considers discrete traits like consenTRAIT and it is less strict than consenTRAIT in testing for a phylogenetic signal. The consen TRAIT can produce more precise result than D-test of Fritz and Purvis, why the authors still used the D-test of Fritz and Purvis?

Response to reviewers' comments

First of all, we would like to express sincere appreciation to the reviewers for their constructive comments. We have carefully revised our manuscript according to their suggestions and as a result, the manuscript is much improved.

Please find detailed responses (in blue) to all of the reviewers' comments attached below.

Reviewer #1:

This work examines the concerted responses of soil bacterial taxa to nitrogen addition across six published studies that each compared a nitrogen amendment to an untreated control. The major finding is that, for many taxa, a taxon's positive or negative response to nitrogen addition is largely consistent across studies and within its clade. This study asserts that these conserved responses will be useful to predict the most likely responders to nitrogen additions by considering the composition of the pre-N addition community. Overall, the study is interesting, and stops just short of moving forward from the observation.

I make two major comments and a few minor ones. Thank you for the opportunity to read this piece. I hope that my comments are helpful to the authors.

Thank you for the feedback, the comments were very helpful in improving the manuscript.

Major comments

1. Bench marking sampling effort. The introduction discusses differences in sampling effort across studies (paragraph starting Line 52) and how this may bias the ability to extrapolate and interpret predictions. It is suggested that understanding the phylogenetic conservation of responses will help to overcome limitations of under-sampling. Despite this emphasis on issues with sampling in the introduction, there are two steps in the methods (pertaining to sampling) that could be benchmarked to assess the influences of those steps on results. These steps should be assessed to determine whether they alter results or interpretation.

Thank you for pointing out these areas of confusion. The reviewer correctly points out that we discuss how "sampling biases" might change the picture of how a taxon response to a perturbation in the introduction. However, we did not mean to invoke "sampling effort" (sequencing depth), but rather differences in other taxa that are present at a location. This variation might be partly due to sampling effort, but probably mostly due to real differences in the presence and abundance of taxa at different locations. We now change this wording in the Introduction (lines 53-61) and Figure 1 to clarify.

Sequence depth

a. First, study does not standardize for observational effort (here, sequencing depth) across the different studies included. The study uses DESeq2 for assessing log₂ changes without prior standardization to an equal sequencing depth across samples. This will allow for some rare members to be observed in samples that have better observational effort (deeper sequencing), and may influence results. I have not noticed any meta-analyses that have used DESeq2 to standardize across studies (that generated data from different sequencers, labs, DNA extraction protocols, etc) but some benchmarking will help the field to understand DESeq2 utility for meta-analyses in general and for consistency in N responses in particular. I suggest performing a direct comparison of resulting log₂-changes across data subsampled to an equal observational effort and data that has not been.

As the reviewer notes, we did not use DESeq2 to standardize across studies. We only used DESeq2 to normalize the sequence numbers by sample within each location. We use this procedure instead of rarefaction within each location.

In terms of comparing across locations, we purposely did not standardize for sampling effort. Rarefaction (or some type of effort standardization) is definitely needed when comparing diversity metrics among samples. However here, we are not comparing samples directly with one another. Instead, we are estimating a treatment response parameter (average log-fold change) of each OTU within a location and testing whether this parameter is consistent across locations. Unlike many diversity metrics, this parameter – log-fold ratio of the control and N addition plots within a location – should not be biased one way or another by sequencing differences across locations, although it will presumably get more accurate with more sequencing. Therefore, after standardizing for sampling effort within a location using DESeq2, we use all sequence data available at each site to estimate this parameter for each OTU at each location. We have now added wording explaining this in the Methods (lines 249-259 and 310-316).

Overall, our goal is to ask whether an OTU's response are phylogenetically consistent **despite** differences in depth of sequencing, sequencing platform, extraction protocols, etc. across locations. There is no way to control for all of these differences. However, the fact that we still significant phylogenetic patterns despite all the potential noise, suggests to us that the pattern is quite strong.

b. Second, the study omits “noise” by removing taxa that were detected in less than half of the plots per study (L241). This step may allow for some responsive taxa with lower occupancy to be not considered across studies, which may also influence results. It will also be more likely to filter rare taxa, which are statistically expected to have lower occupancy. However, including all observed taxa is one of the advantages of DESeq2 – that one doesn't have to “throw out” high-quality data despite differences in sequencing effort across samples. What is the benefit of this step for the log₂-change results and how does omitting it change outcomes?

c. I also wonder about the consequences of removing taxa with less than 50% occupancy prior to the DESeq2 log₂ change analysis – my understanding was that the shrinkage approach

of DESeq2 requires all data, including the noisy points, to best estimate the variance structure before calculating log₂ fold changes?

Thank you for bringing up this point. We now clarify (lines 251-259) that we use DESeq2 in a limited way that differs from how it is typically used for RNAseq data. As the reviewer mentioned, the “noise” or low-frequency genes can be used to determine the statistical significance of the responses in the DESeq2 program. While we used DESeq2 to calculate normalized log-fold responses, we did not use it to determine the statistical significance, so the removal of the locally rare OTUs does not affect our results. Instead, we exported the response ratios for phylogenetic and statistical analyses in other programs. (Using DESeq2 to determine significance does not work for our purposes, because the number of significantly responding OTUs at any one site is very small relative to total diversity and there is almost no overlap of significantly responding OTUs across sites.)

Sequence depth

d. Information can be added to the main text, like QC sequencing depth across studies (in addition to the total number of OTUs observed, number and taxonomy of OTUs excluded at the occupancy filter step, to help readers interpret the range and variation in the studies included and its consequences for cross-study comparisons.

Thank you for suggesting this. We now include a supplementary table (Table S1) that presents the source of sequence dataset, the number of quality-filtered sequences, and the number of OTUs before and after removing locally-rare OTUs.

Functions consistently responding taxa have

2. Moving towards mechanisms. The study ends with a conclusion of consistent N responses across studies and conserved within some major lineages. It is a brief piece that stops short of investigating the “why”. The next step is to ask what functions these lineages have that support their consistent responses. The article would be enhanced if there were some discussion, or, better, some investigation, of the functional genes related to nitrogen responses/cycling that are conserved within the responsive lineages (L117). There are genomic resources available for many of the responsive clades – the authors could consider some representative comparative genomic cases, perhaps of the most phylogenetically conserved and maximally responsive lineages, to explore their genomes for conserved functional genes that would explain their responses. They could also test their complexity hypothesis of related genes (discussion starting L150).

We agree with the reviewer that linking these patterns to a mechanism would be an excellent next step, and we now add some discussion about this direction (lines 153-166). Unfortunately, it is unclear what genetic signatures might be related to a bacterium’s nitrogen response. The ability to assimilate nitrogen is one obvious trait. However, recently our colleagues surveyed the publicly available bacterial genomes and show that the vast majority (93%) of sequenced bacteria possess the genes for ammonia assimilation, so the presence/absence of these genes will not help differentiate among taxa or clades (Albright et al. 2018 *Microbial Ecology*). Thus,

we hypothesize that the response will depend not only on N assimilation ability (or other N cycling pathways), but a mixture of traits, including nutrient transporters, the ability to grow quickly when N availability suddenly increases, and differences in the nutrient requirements of the cell. Furthermore, even with currently available genomic resources, there is little overlap between sequenced genomes and soil bacterial OTUs. In a recent analysis, only 1.4% of all Earth Microbiome Project diversity shared >97% similarity to the 16S genes from available genomes (Choi et al. 2017 *ISME* 11: 829-834). Overall, an investigation of these mechanisms will not be easy and therefore, seems beyond the scope of this paper.

Minor comments

L86 – a bit unclear as to whether the entire set of analyses were performed at the 97% OTU level or just the cross-location analyses? I'm not sure if the exact sequence variants comment is needed if these are not ever used.

The entire set of analyses was performed with the OTUs clustered at 97% sequence similarity. To avoid confusion, we added the flow diagram of methodology as a supplementary figure (Fig. S2). We kept in this comment about ESVs, however, because other related studies have used ESVs. We add a citation here to clarify.

Were there any taxon/lineage responses conserved within nitrogen level or type (as reported in Table 1)? One hypothesis is that that type of nitrogen would influence taxa that directly respond to particular substrate but potentially not indirectly responsive taxa.

This is a great question. At this point, we cannot test how nitrogen level or type affects our results, as the number of studies available to us is limited. As Table 1 shows, N addition level (8-160 kg-N/ha/yr) and N addition type (urea or NH_4NO_3) differ by location. Our current analysis rests on identifying many taxa that are found at least in 5 sites, allowing us to test the consistency of an individual's response. Very few taxa would reach this criterion if we grouped the sites into separate categories.

Though Figure 1 provides a nice visualization for the types of patterns that could be observed, some aspects of it are unclear. Panel A dendrograms demonstrate potential different patterns (under and over dispersion) within a particular location, which is not directly addressed in the main text. Panel B is comparing phylogenies across two locations, and also builds into the protocol in Panel D and so may not be needed as a separate panel. Panel C is representing the sampling issue, but is not directly addressed in the paper with analysis (though bench marking as described above would help this). Panel D summarizes the study at hand. Overall, I think this figure could be removed. If the authors want to keep it, please clarify these parts and consider moving it to supporting materials.

We agree that this figure was confusing and thank the reviewer for pointing this out. We have kept the same ordering of the panels to parallel the flow of the Introduction text, but now add text to the figure to clarify the different points we were trying to make. We have further clarified the explanation of Panel C in the text and in the figure legend (lines 447-456).

Figure 2A – the Green trace (Leff_1?) is an obvious outlier. Why?

We assume the reviewer is referring to the two significant points are 0.21 and 0.22 that are negative. These are caused by opposing responses across deeper genetic clades. We now mention this in the Results (lines 112-113).

Figure 3 – the authors describe lineages that have conserved responses at different resolutions, but this tree is too crowded to discern some of those specific discussion points. I'm not sure that the tree is helpful.

We understand that Figure 3 only allows visualization at the phylum level. However, it seems important to include, because it gives an overall sense of the scale of phylogenetic conservation. At the same time, we referred to Table 3 for the details that we describe. We have now moved the information in Table 3 to Figure 4 to make these data easy to follow.

Tables 1 and 2 – I wonder if these could instead be represented as figures? For example, as bars centered around zero, with increases presented as positive bars, and decreases as negative? As it stands, it is hard to digest the range and variability in the responses by reading such a large table, and it is also hard to “ignore” the non-significant changes that are presented in regular (not bold) font. The Table legends should clearly state how the results were derived. I think that Table 1 is related to the output from DESeq2, but then I am confused as to how the percentages in change are calculated, because I thought that the test considered log2-fold change, so shouldn't the change be represented as log2-fold rather than %? Similar comment for Table 2. If results are not significant, I suggest removing values and just stating “n.s.” Has some kind of multiple comparison adjustment been performed?

It seems to us that the Reviewer must have been referring to Tables 2 and 3.

We understand the reviewers comment about removing the n.s. values in Table 2, however, we have chosen to keep them here for a very specific reason. In this instance, the tests for the significance of the clustering of positive and negative responses are not independent. If positive responses to N are clustered on the phylogeny, then the negative responses are also very likely clustered. For this reason, it is useful to see that the values for the depth of conservation are very similar, even if the specific test may not be significant.

For Table 3, we thank the reviewer for the impetus to turn this into a figure. This information is now in Figure 4. The values (percent) in Table 3 (now Figure 4) are not derived directly from the DESeq2 analysis. We now remind the reader how the values are derived in the figure legend (lines 470-475). The bar graphs show the percent of OTUs with positive or negative responses by taxonomic group.

For Table 2 - it strikes me how there is not much of a range in genetic depth (TD), even among statistically insignificant values, though the original 2013 paper suggests that ~0.17 is a

relatively high degree of phylogenetic conservation – it is so high that it is not within the range of the data shown therein (Fig 2). It would be interesting to see the distribution of these values – is there some expected range or variability of TD given the number of branches on the phylogeny?

Yes, it is quite surprising how consistent the TD values are! This is one of our main results, which suggests that the depth of conservation is not context dependent (or inaccurate because of which taxa are detected). As requested, we plot the distribution of genetic depth of the phylogenetic clades in Figure S6. We are not quite sure what the reviewer is referring to in the original paper, as ~0.17 would be extreme. The original paper reports that 0.10 is the mean depth for oxygenic photosynthesis, the deepest trait considered.

Fig S4 – it looks like the Wang sample is missing?

Response: Thank you for pointing out our mistake. We have added Wang to Figure S5.

Throughout, I was wondering if the percent change/divergence was referring to the full 16S rRNA gene or the amplicon (e.g., L 99, L 140, L162 and elsewhere).

The percent divergence refers to the partial 16S rRNA amplicon sequence. We revised these sentences (lines 100, 147, and 141).

L203 Inclusion criteria. This study has focused on native ecosystems like grasslands and bamboo forest, but is the N response consistent and conserved also in agricultural systems? Inclusion of agricultural manipulations would increase the study size and, potentially, the generality of the conclusions.

We agree this would have been very interesting! We included two croplands (O'Brien and Wang) in the analysis as shown in Table 1. We tried to include as many studies as possible, but unfortunately the sequence dataset and/or environmental metadata were not available (even after bugging the authors several times) for many studies that might have been useful (lines 214-222).

L 228 singleton sequences or OTUs? I interpret this as sequences.

Yes, we used the singleton sequences. We revised the sentence (L238).

Share the workflow

A computing workflow (including sequence processing through to statistics in R) should accompany the work so that others can use and replicate the analysis. The metadata curated for this particular meta-analysis should be made available, as should the merged OTU table and the representative OTU sequences used to make the phylogenies because these are needed to reproduce the analyses (L315).

We are delighted to share the reanalysis of these datasets including OTU tables with assigned taxonomy (before and after removing rare OTUs), the results of DESeq2 (response of OTUs), OTU sequences, and phylogenetic trees. We have archived these on a GitHub site and included this statement at the end of the article (data availability section).

An interesting aspect of this work is that it is considering and analyzing N responses as if an increase or decrease after N amendment were in itself a particular microbial trait, but is it? As a contrast, nitrogen fixation is a bacterial trait encoded by a particular set of functional genes. It would enhance the work to include a brief discussion of responses v. traits, their overlap, and their distinctions, and how to use both to move forward toward predictions...

Thank you to the reviewer for bringing up this salient point, which we agree deserves a more straightforward discussion in the manuscript. We avoid calling the response itself a trait, but prefer to consider the traits that determine this response – so-called response traits (*sensu* Laval and Garnier 2002 *Functional Ecology*). Thus, these response traits influence a species' response, a change in abundance quantified by a "Specific Response Function" or "Process Response Curve" (Suding et al. 2008 *Global Change Biology*; Allison and Martiny 2008 *PNAS*). We now clarify this point (lines 147-150).

...If the authors include an analysis of functional genes conserved for N utilization within responsive lineages, this would allow them to potentially directly link responses to genetically-encoded traits.

Please see our response on page 4.

Reviewer #2:

The authors present a conceptual figure to help follow the line of argument developed in the introduction. I generally understand (and like) the idea of having this figure, but it has several unclear elements and, so at the end it generates confusion. I particularly don't understand how local and sampling effects are represented in scenarios B to D.

We are sorry the figure was not clear. As we describe above, we have added text and labels to the figure to clarify and we reworded the explanations in the Introduction. We hope this figure now makes sense.

The authors should also justify why they exclude from their framework and several subsequent analyses those taxa that are not responsive to N additions. The lack of response in this case can be indeed very informative, and all locations and phyla contain many non-responsive OTUs (according to Fig. S2). Linked to this comment, I suggest that the magnitude of phylogenetic conservatism is tested after coding bacterial responses to N addition as a continuous trait (direction and magnitude of change, including 'no change' values) instead of coding the response as binary trait (positive vs negative) (and using e.g. Blomberg's K instead of D statistics).

We apologize for the confusion here. We do not exclude any taxa other than very rare OTUs for which we cannot estimate a reliable response ratio. We classified all taxa as either of positively or negatively responding based on the response ratios calculated in DESeq2 (even if the response was very close to zero). Just to be clear, we now also clarify in the legends of Figures 3 and S4 that the black branches are **not** non-responding OTUs, but that we are only colored the consensus clades. In terms of considering the response as a continuous trait, we performed the mantel correlogram analysis to address this point. We now reiterate this in Results sections (lines 107-113). However, one limitation of using continuous values (in the Mantel correlogram or other metric such as Blomberg's K) is that we cannot identify discrete clades without defining a discrete cutoff. It therefore seems to us that the most natural cutoff is whether the values are positive or negative.

More information is needed in the materials and methods section to allow an adequate evaluation and future reproducibility of the study. This point is particularly relevant as regards two main questions.

First, sequencing depth and OTU table construction: it is unclear whether the authors subsampled sequences at the same level in all studies and whether OTU tables represent number of counts or relative abundance. It is essential to justify this issue as most phylogenetic analyses are based on the positive (or negative) OTU response to N addition. In case OTU tables reflect relative abundances, the increase in some OTUs is necessarily accompanied by the decrease in other OTUs.

As described in response to Reviewer #1, we now clarify these points in the Methods section (lines 251-259). We also acknowledge the issue of relative abundances in the Introduction and Discussion (lines 49-52), as this is a key reason why we might not expect that the responses would be consistent across locations.

Second, more details and analyses are needed as regards the phylogenetic reconstruction. Using short reads to reconstruct phylogenies has several drawbacks that remarkably reduce tree reproducibility (often leading e.g. to many misplaced sequences even at the phylum level). I'd suggest that the authors recalculate all phylogenetic signals in several independently built trees to show that results are reproducible. When checking the consistency of their results, I'd suggest using the more robust maximum likelihood (instead of neighbor joining) trees.

We agree that short reads are not good for phylogenetic reconstruction. That said, our phylogenies are remarkably consistent with the independent taxonomic classification of the reads. In particular, the outer ring of Figure 3 shows that the phylum classifications cluster as expected (the exceptions are likely due to incorrect database annotations).

We further appreciate the Reviewer's concerns about the reproducibility of the trees, however, we respectfully submit that we have a unique opportunity here that differs from many studies. Most studies focus on estimating reproducibility within a site (e.g., bootstrapping), because they don't have the ability to actual reproduce the study. Here, however, we can reproduce our

results in 13 locations. Therefore, we focus on reporting the variability in our estimates across locations (Table 2).

It is also a reasonable question to wonder if our results depend on our choice of NJ versus ML trees. We used NJ trees because the genetic scale of these trees are more aligned with the way microbial ecologists think of genetic depth (i.e., in terms of genetic dissimilarity of 16S sequences). The depth in maximum likelihood trees are not easily interpretable in this way, so it does not make sense to report the TauD values. However, the general conclusion that N response is phylogenetic conserved does not depend on the reconstruction method. To show this, we now report the correlation between the genetic distances of the OTUs from the two trees. The branch lengths between OTUs in NJ and ML trees were highly correlated for the trees within a location (mean $r = 0.84$, range 0.75-.90, all $p < 0.001$) and for the cross-location tree ($r = 0.83$, $p < 0.001$)(Table S3). Further, the consenTRAIT metric using ML trees was significant in almost an identical pattern as observed for the NJ method. We have added these analyses to the Methods (lines 284-288) and the Results (lines 104-106; Table S2).

Please, find more details below on information that should be added to materials and methods. Finally, the argumental flow in the introduction is clear and reasonable. However, many statements are not properly referenced based on published literature. Please, find some suggestions below.

Other comments

L20. five continents according to the main text and fig. S1
We fixed (line 20).

L26. Typo (add a full stop after 'location')
We fixed (line 26).

L39-41. This would be a good place to explicitly introduce the differential depth of conservation of bacterial traits, that you later analyse and discuss
We revised as suggested (lines 40-41).

L43-59. Please, support your arguments about local effects on published literature.
We have added some citations here. However, several of these statements are our own hypotheses and as far as we know, have not been tested (lines 47-52).

L66-76. I'd move the paragraph on N addition to the first place in the introduction (this is a matter of taste, but I find it out of place here)
We appreciate the idea and have gone back and forth on this very decision. However, we have left the current structure to emphasize that our main focus is the phylogenetic conservation of responses to environment change, whereas N addition is the specific test of this question.

L75-76. Alternatively, could it indicate that the response is phylogenetically conserved at shallower clade depths?

Good point. We have revised as suggested (line 78)

L204-211. It would be a good idea represent the selection in a Prisma Flow Diagram in Supplementary Information (not necessary, but informative).

Thank you for this suggestion. We have added a flow diagram of the methods (Figure S2).

L215-234. The authors should make an effort to provide more detailed information. I'd suggest: For published studies: sequence availability, sequencing technology (from L222 I understand that not all studies used Illumina, but I'm not sure), sequencing depth, treatment of different levels of N addition (did you somehow differentiate between N dose?), how did you treat replicated plots within locations?

We have created a supplementary table with this information (Table S1). We also clarify these sentences in the method section (lines 251-254). The sequencing platform of all studies was Illumina Miseq.

For sequence processing: tool used for filtering chimera, reference SILVA database, sequencing depth used for OTU construction (did you standardize across studies?), content of OTU tables (do they reflect number of counts or relative abundance?).

We have revised the sentences as suggested (lines 235-246). We used the scaling approach for normalization of sequence depth within locations as described above.

For phylogeny reconstruction: please, give the phylogenetic trees and further details (e.g. did you align or remove hypervariable regions?)

We provide the phylogenetic trees in Fig S4. We aligned the sequence of the V4 region of 16S rRNA gene (lines 262-264).

L256. Could you add a third level (i.e. no response)? Otherwise, I'd change 'discrete' to 'binary'. As described above, the OTUs were only classified into positive and negative responses. We changed 'discrete' to 'binary'.

Table 1. What do asterisks mean in the column 'Replicate of plots'?

We revised the caption (Table 1).

Fig1. Please, indicate in the caption what do left and right trees represent in scenarios B and C. How are locations represented in B. Also, I don't find scenario D very informative.

We revised Figure 1 by adding text to clarify.

Fig2A. Maybe if dots were smaller, the figure would be easier to read.

We revised the figure as suggested (Fig. 2A).

Table S1. What does the 'Mean' reflect in this case?

We changed to "Mean of each location" (Table S2).

Figure S3. Could you make coloured bars showing the phylum thicker?

We revised the figure as suggested (now Fig. S4).

Reviewer #3:

(1) As we know, the duration and intensity of N addition will affect the extent of microbial response. In Table 1, the authors listed the level/type of N addition in all the 13 sampling sites and the amount of N addition ranges from 8 to 160 kg N/ha/yr. This is a wide range of N addition, which can cause different levels of response. For the taxa with relatively wide ecological niche, they may not respond to 8 kg N/ha/yr N addition but respond to 160 kg N/ha/yr N addition. Is it acceptable to explore the microbial response to N addition without considering ~20-fold difference in N addition amount? Besides, the duration and the timing of N addition is also important to understand microbial response to environment perturbations, but I can't find the related information.

(2) Another important point needed to be considered is the N nutrient status in the original sampling sites. If the sampling site is N-limited, we may detect more response, while if the sampling site is N-saturated, the response might be different. The lack of the ambient N condition in soils and ambient N deposition in each sampling site makes it hard to understand the findings in this study.

We understand the concerns the reviewer proposed. Indeed, as we explain in the Introduction (lines 44-47), we hypothesized that we might not see consistent responses across locations in part because of variation in the N experiments and the baseline environments (Figure 1B). Thus, the fact that we do see a consistent response suggests that the context of the locations and experiments did not overshadow the phylogenetic signal (lines 137-139). This was surprising to us, too! If this variation did matter, we would expect to see little or no phylogenetic signal when the datasets were merged across locations.

(3) The “global scale” is a little exaggerated as there are only 13 sampling sites included in the study. I feel 13 sites are not enough to produce robust results using meta-analysis. We removed the term “global scale.” We also wish we had more sites, but these were the only ones we could find that matched our selection criteria (Figure S2). However, given the robust results with only 13 sites, we would argue that the study is still worth publishing. We also hope that our framework can be helpful to others in future analyses – hopefully with more study sites!

(4) line 26, should be “across location. Our findings”.
We fixed (line 26).

(5) line 78-81, question 2 and question 3 seem a little repeated.
The two questions are indeed parallel (within locations versus across locations), but we address them in separate sections in the Results. Hopefully, the revised Figure 1 helps to clarify.

(6) line 94-95, Figure S2 showed the changes of bacterial composition to N addition in phylum level rather than in OTU level.

Figure S2 (now Figure S3) shows the response of individual OTUs (each point), organized by phylum. We now clarify in the figure legend (line 482).

(7) line 269-272. D-test of Fritz and Purvis considers discrete traits like consenTRAIT and it is less strict than consenTRAIT in testing for a phylogenetic signal. The consen TRAIT can produce more precise result than D-test of Fritz and Purvis, why the authors still used the D-test of Fritz and Purvis?

We also prefer consenTRAIT! However, the two tests are different, and many researchers like to see a more traditional statistic. The D-test tests for phylogenetic conservation while assuming Brownian motion of vertical evolution. ConsenTRAIT tests for conservation against a random distribution across the tree, which would be expected with high levels of horizontal gene transfer.

Reviewers' Comments:

Reviewer #1:

Remarks to the Author:

The revisions to the manuscript have addressed my major concerns. Thanks to the authors for their clarifications and efforts. I think piece this will be a good contribution to the journal.

Reviewer #2:

Remarks to the Author:

It has been a pleasure to read the revised version of this manuscript. The authors have appropriately addressed all my previous comments. Please check completeness of merged tree in Fig. 1D, and typo in L190.

Reviewer #3:

Remarks to the Author:

The authors basically solved the reviewers' concerns and greatly improved the MS. Minor comments:

(1) Line 85: "across 4 continents" should be 5 continents.

(2) Line 91-92: why the author emphasized the broader definition of taxa is better? It's a little confused. It's better to explain why 97% sequence similarity is better than ESV? Or delete the description.

(3) How many non-response OTUs were removed for phylogenetic conservation analysis, which should be clarified in the Methods or Results parts.

Response to reviewers' comments

First of all, we would like to express sincere appreciation to the reviewers for their constructive comments. Please find detailed responses (in blue) to all of the reviewers' comments attached below.

Reviewer #1 (Remarks to the Author):

The revisions to the manuscript have addressed my major concerns. Thanks to the authors for their clarifications and efforts. I think piece this will be a good contribution to the journal.

Reviewer #2 (Remarks to the Author):

It has been a pleasure to read the revised version of this manuscript. The authors have appropriately addressed all my previous comments. Please check completeness of merged tree in Fig. 1D, and typo in L190.

We checked and fixed the typo

Reviewer #3 (Remarks to the Author):

The authors basically solved the reviewers' concerns and greatly improved the MS. Minor comments:

(1) Line 85: "across 4 continents" should be 5 continents.

We fixed (line 90).

(2) Line 91-92: why the author emphasized the broader definition of taxa is better? It's a little confused. It's better to explain why 97% sequence similarity is better than ESV? Or delete the description.

To test the consistency of response across locations, we needed to compare the response of the same taxa at many locations. As we describe (lines 95-96), the broader definition of taxa (rather than exact sequence variants) makes the comparison more feasible.

(3) How many non-response OTUs were removed for phylogenetic conservation analysis, which should be clarified in the Methods or Results parts.

We received the same comment from Reviewer #2 and have revised the method section (lines 257-259, 272-273). We do not exclude any taxa other than very rare OTUs for which we cannot estimate a reliable response ratio. We classified all taxa as either of positively or negatively responding based on the response ratios calculated in DESeq2 (even if the response was very close to zero).